# Bactericidal Effect of Clove Oil against Multidrug-Resistant *Streptococcus suis* Isolated from Human Patients and Slaughtered Pigs

**DOI:** 10.3390/pathogens9010014

**Published:** 2019-12-21

**Authors:** Kanruethai Wongsawan, Wasana Chaisri, Sahatchai Tangtrongsup, Raktham Mektrirat

**Affiliations:** 1Department of Veterinary Biosciences and Public Health, Faculty of Veterinary Medicine, Chiang Mai University, Chiang Mai 50200, Thailand; KanreuthaiW@gmail.com; 2Department of Food Animal Clinics, Faculty of Veterinary Medicine, Chiang Mai University, Chiang Mai 50200, Thailand; wasana.ch@cmu.ac.th; 3Department of Companion Animal and Wildlife Clinic, Faculty of Veterinary Medicine, Chiang Mai University, Chiang Mai 50200, Thailand; sahatchai.t@cmu.ac.th; 4Research Center of Producing and Development of Products and Innovations for Animal Health and Production, Chiang Mai University, Chiang Mai 50200, Thailand; 5Research Center of Pharmaceutical Nanotechnology, Chiang Mai University, Chiang Mai 50200, Thailand

**Keywords:** clove oil, *Streptococcus suis*, antibacterial activity, bactericidal activity

## Abstract

*Streptococcus suis* is a zoonotic pathogen that is currently considered an emerging multidrug-resistant (MDR). Increasing antibiotic resistance can lead to the unsuccessful treatment of *S. suis* infection. Recently, many investigations of medicinal plants were conducted for the treatment of infection as a result of the increase of antibiotic-resistant bacteria. The aims of this study were to determine the chemical composition of essential oil from *Syzygium aromaticum* (L.) Merr. & L.M. Perry and the antibacterial activities of clove oil on MDR *S. suis*. Using gas chromatography coupled to a mass spectrometer, eugenol (97.76%) was found to be the major active ingredient of clove oil. *In vitro* antibacterial activities of clove oil against MDR *S. suis* were evaluated. Using the agar disc diffusion test, the clove oil showed a maximum zone of inhibition at 15% (v/v) oil concentration. In a broth microdilution method, the minimum bactericidal concentration of clove oil against all MDR *S. suis* isolates was 0.1% (v/v). A time-kill analysis was performed, and the killing kinetics of clove oil showed that MDR *S. suis* was completely reduced after 15 min of exposure to clove oil. In addition, clove oil exhibited a strong antibacterial activity at all pH values applied following incubation of MDR *S. suis* in pH-adjusted media with clove oil. Moreover, scanning electron microscopy revealed the nonviable *S. suis* isolates clearly showed atypical form and cell membrane lysis after incubation with clove oil. This study confirms the efficacy of clove oil as a natural antimicrobial against MDR *S. suis* and suggests the possibility of employing it as a promising alternative product for control of infectious diseases caused by *S. suis* in animal and human patients.

## 1. Introduction

*Streptococcus suis* is an emerging zoonotic pathogen that causes severe systemic infection in humans who come in close contact with pigs or pork-derived products. It also causes economic losses in the intensive swine industry. *S*. *suis* infections have been reported worldwide and a high human morbidity rate has been observed in Thailand, with the majority of infections caused by the consumption of raw pork [1].

The use of antimicrobial agents in veterinary medicine is essential to control infectious diseases. However, the development of resistance of *S. suis* has been increasingly reported [2,3,4,5]. The resistance of *S. suis* to antimicrobials commonly used in swine, including lincosamides (lincomycin and clindamycin), macrolides (erythromycin, spiramycin, and tylosin), sulfonamides, oxytetracycline, and tetracycline, has been acknowledged in many countries [4,6,7,8]. In human strains, resistance to tetracycline and macrolides has been reported since 2000 [9], and increased resistance rate has recently been reported in Asia [6,8,10,11,12,13,14]. Penicillin and cephalosporins have been considered the primary drugs of choice for *S. suis* infection. However, resistance to these drugs has been reported in many countries, including Thailand [4,11,12,13]. 

Thailand has the highest antibiotic resistance rates (with 100% lincomycin and clindamycin resistance, and 90% tetracycline and gentamycin resistance) compared with other Asian countries, and has the highest rate of penicillin resistance compared with other continents [13]. Currently, *S. suis* is considered a newly emerging multidrug-resistant (MDR) zoonotic pathogen [15,16,17]. This could lead to an unsuccessful treatment of *S. suis* infection and become a major public health concern. Thus, finding new and alternative antimicrobial compounds for the treatment of infectious diseases from medicinal plants is of particular interest.

During the last decade, medicinal plant investigations for the treatment of infectious disease have intensified due to the increase of antibiotic-resistant bacteria. Clove oil is extracted from the dried flower buds of clove (*Syzygium aromaticum* (L.) Merr. & L.M. Perry). It has been used as a perfume and as a food flavoring [18]. Medicinally, it is widely used for reliving toothaches or cavity problems, asthma, rheumatoid arthritis, acne, scars, various allergic disorders, and as an antiseptic in oral infections [19,20]. Moreover, the antimicrobial properties of clove oil and its application to any product, such as food and health products, have been tested and have shown inhibitory activity on various pathogens, including *Listeria monocytogenes*, *Campylobacter jejuni*, *Salmonella enteritidis*, *Bacillus cereus*, *Escherichia coli*, and *Staphylococcus aureus* [21,22,23,24,25].

Although the antibacterial properties of clove oil have been previously reported, the study of its effect on MDR *S. suis* has been limitedly examined. The aims of this study were to determine the chemical composition of essential oil from clove buds and to investigate the antibacterial effects of clove oils on MDR *S. suis*. Our study revealed a strong potential bactericidal effect of clove oil against the swine pathogen MDR *S. suis*. 

## 2. Results

### 2.1. Chemical Compositions of Essential Oil

The clove oil appeared as a clear pale yellowish liquid. The phytoconstituents were characterized by a GC-MS method with a running time of 55 min. The chromatogram showed the presence of four identifiable spectra (Figure 1). A list of the constituents identified in the clove oil and their percentage composition are shown in Table 1. The results demonstrated that the clove oil was composed of eugenol as the major component (97.76%). Caryophyllene (1.64%), gamma-humulene (0.39%), and caryophyllene oxide (0.78%) were observed as minor constituents.

### 2.2. Antibacterial Activity of Clove Oil

Clove oil was found effective against all of the tested MDR *S. suis* isolates, as summarized in Table 2. The diameter of inhibition zones depended on the concentration of the oil. Maximum inhibition was found at 15% concentration, and the minimum effect was observed at 5% concentration. The inhibition zones were 5.07 ± 0.12 to 13.07 ± 0.12 mm, 5.03 ± 0.05 to 14.03 ± 0.05 mm, and 8.00 ± 0.00 to 15.07 ± 0.05 mm on discs impregnated with 5%, 10%, and 15% (v/v) oil, respectively. At 1% (v/v), the oil could not inhibit any the tested isolates. As a positive control, high sensitivity to penicillin (10 µg) was observed for all MDR *S. suis* strains. No inhibition zone was observed in the discs with methanol. This indicates that the methanol used to prepare the clove oil solution had no effect on bacterial growth.

### 2.3. Determination of MIC and MBC

The minimum inhibitory concentration (MIC) and minimum bactericidal concentration (MBC) results are summarized in Table 2 and the distribution of MICs and MBCs for 15 isolates is shown in Table 3. The results showed that the MDR *S. suis* isolates were inhibited by clove oil at a very low concentration, with the MIC_50_ value at 0.05% and the MIC_90_ value at 0.1%. The clove oil showed bactericidal effect against all tested MDR *S. suis* with the MBC values were 0.1 ± 0.00 %. In addition, as a vehicle control, methanol was checked for its antibacterial capability at the various concentrations applied for preparing the essential oil in the experiment. We found that methanol at each concentration did not affect the growth of the tested isolates (data not shown).

### 2.4. Time-Kill Study

To determine the killing kinetics of clove oil against MDR *S. suis*, survival of the MNCM06 isolate was evaluated over a 24 h period in the presence of clove oil at MBC. The data on CFU showed that treatment with 0.1% (v/v) essential oil exhibited a strong bactericidal effect on the MNCM06 isolate in a rapid and time-dependent manner. The cell population was reduced by approximately 2-log CFU/mL after only 5 min of incubation. Within 15 min, the bacterial population was completely inactivated (Figure 2, Appendix A). Loss of viability was apparent for the MNCM06 isolate after prolonged exposure to the clove oil, while the growth rate was not decreased for the control without clove oil. 

### 2.5. Effect of pH on Bactericidal Activity

The highest bactericidal activity of clove oil was observed at pH 4. However, the bactericidal effect at the same time of treatment were not significantly different (*p* > 0.05) among pH values applied (Figure 3, Appendix A). The results show that at an MBC of 0.1%, clove oil completely inhibited the tested isolate within 15 min of incubation, with a strong activity at pH 4 through pH 8. The growth of the tested isolate in the control tube without clove oil was unaffected by changes in pH value.

### 2.6. Effect of Clove Oil on Cell Surface of S. suis

The morphological changes of the treated bacteria were compared with the untreated control, as shown in Figure 4. The cell surface of the untreated MNCM06 isolate was found to have typical morphology, which is an intact spherical shape (Figure 4a). In contrast, the treatment of cells with clove oil for 1 h induced important morphological damage (Figure 4b–d). At the MIC and MBC of clove oil, it was found that cells became deformed, with cell wall ruptures and cell lysis in which leakage of cytoplasmic contents were apparent. 

## 3. Discussion

Four phytoconstituents were identified in essential oil from clove buds in this study, and eugenol (97.76%) was the major constituent. This is in agreement with previous studies that reported eugenol was the main component of clove [22,26,27]. However, the percentage of eugenol in those studies was different, ranging from 50.3% to 76.8%. Our study obtained a higher eugenol content and found different components in the clove oil. The differences in the contents and components of the essential oil from clove buds can be explained as influences of the clove species, geographic region, environmental factors (temperature, humidity, and light) [28], and the difference in extraction methods used [27,28,29]. More importantly, the high concentration of eugenol in clove oil gives it potent bactericidal properties [30]. Moreover, the synergistic effects of phytochemicals are well-documented [31]. Interestingly, previous studies showed that the MIC of the purified eugenol is higher than the total clove oil [32]. Therefore, the antibacterial activity of clove is due to phytocomplexes of eugenol and the minor constituents.

The present study was performed on *S. suis* isolates, which are resistant to many antibiotics that are widely used to treat bacterial infections and used as feed additive in pigs to prevent infections in Thailand. The results from the agar disc diffusion method revealed that clove oil had inhibitory effects against all test MDR *S. suis* isolates. This is consistent with previous studies [33,34,35,36]. Incidentally, regarding the inhibition zones obtained from the essential oil, it is notable that differences can occur among studies due to the methods used in solubilizing the oils to obtain hydrophilic molecules [33], antimicrobial assays, and tested organisms. Even though the antibacterial properties of essential oil from clove bud against *S. suis* have already been reported [37], its effect against MDR *S. suis* has not been previously reported. Our findings suggest that the MDR *S. suis* isolated from both human patients and slaughtered pigs is susceptible to the essential oil of *S. aromaticum* (L) Merr. & L.M.Perry.

The bacteriostatic and bactericidal activities of clove oil were assessed by determining the MIC and MBC, respectively. The MIC_90_ and MBC_90_ values were 0.1%, which indicates that the action of clove oil on MDR *S. suis* is bactericidal. The MIC/MBC values of clove oil obtained in this study were lower than in a previous study by Perugini Biasi-Garbin et al. [38], which reported 0.125 to 0.5% (v/v), while Baskaran et al. [39] reported 0.4/0.8% (v/v) on mastitis pathogen *S. agalactiae*. However, it should be noted that the initial bacterial load, culture medium used, incubation time, and temperature are important variables that can affect MIC determination of clove oil [40,41,42,43].

The time-kill study demonstrated rapid inhibition of bacterial growth within only 5 min of treatment with clove oil at MBC concentration (0.1%). It was found that the clove oil reached maximal bactericidal effect within 15 min of incubation, when a complete loss of the viability of the MDR *S. suis* was observed. Our results are in accordance with a previous finding that the maximum kill rate of *S. pneumoniae* was observed during the first 15 min of eugenol exposure at MBC (0.12%) concentration [44] .However, it seems that *S. suis* was completely killed more rapidly than *S. pneumoniae*, which was completely killed within 60 min of exposure. This result suggests that clove oil has strong bactericidal activity on MDR *S. suis* and also indicates that MDR *S. suis* is very susceptible to clove oil.

With regard to the action of clove oil on MDR *S. suis*, the SEM images showed irregular morphology of cells treated with clove oil compared to untreated cells. Our results showed that cell breakdown is the principal phenomenon for MDR *S. suis* treated with clove oil. This corresponds to the result of the time-kill study, which demonstrated a progressive reduction in the number of viable *S. suis* cells in medium with clove oil and eventually led to complete death of the cells within 15 min. Our findings on the action of clove oil on bacterial morphology are consistent with other studies [22,23,38,43,44,45,46]. It is already known from many research reports that the antibacterial activity of clove oil is due to its major component, eugenol. Eugenol is a phenolic structure that is highly active against various microorganisms [45]. It can readily pass through gram-positive bacterial cell walls, causing degradation of the cell wall [44], and then damage the cytoplasmic membrane, causing destruction of membrane proteins, impairment of bacterial enzyme systems, increased permeability leading to leakage of the cell contents, and eventually cell lysis [22,23].

Moreover, we described the results for the effect of pH on the bactericidal activity of clove oil against MDR *S. suis*, which showed strong antibacterial activity at all pH values applied, with the highest antibacterial activity observed at pH 4. Our result suggested that the bactericidal activity of clove oil against MDR *S. suis* is sustainable through neutral, acidic, and alkaline conditions. However, the variation of pH effect on the bactericidal activity of clove oil was reported in a previous study that found the highest activity of eugenol against *Salmonella typhi* at pH 9.0 [23], while Hoque et al. [47] reported the highest activity of clove oil against *L. monocytogenes* at pH 7.0, and Knight et al. [48] showed strong inhibition of the growth of *E. coli* O157: H7 at neutral (7.2) and acidic (4.5) pH values. The difference among these results could be due to factors such as test organisms, sample preparation, and the method used in pH assays. However, it is notable that the bactericidal effect of clove oil against MDR *S. suis* was not markedly inactivated by changes in pH, which makes it valuable and possible to apply as a treatment or preventive therapy for *S. suis* infection.

In conclusion, this study found that clove oil exhibits a strong potential bactericidal effect on MDR *S. suis*, which suggests the possibility of employing it in a promising alternative product for control of infectious diseases caused by *S. suis*. Further research with *in vivo* studies and clinical trials need to evaluate the potential of clove oil in medical applications. 

## 4. Materials and Methods

This study was performed at the Department of Veterinary Biosciences and Public Health, Faculty of Veterinary Medicine, Chiang Mai University, Thailand. The ethical approval by the Ethics Committee of the Faculty of Veterinary Medicine, Chiang Mai University, Thailand was waived since this study did not involve the study of human or animal subjects, data, or tissues. 

### 4.1. Bacteria and Culture Conditions

The MDR *S. suis* isolates from storage at −20 °C were used in the present study (Table 4). All of them were resistant to tylosin, enrofloxacin, tetracycline, trimethoprim-sulfamethoxazole, and oxytetracycline, which previously characterized antimicrobial resistance based on a breakpoint disc diffusion assay and a broth microdilution assay according to Clinical and Laboratory Standards Institute (CLSI) guidelines [49]. The bacterial strains were recovered from the bacterial storage medium by subculture on Columbia blood agar (Oxoid, Hampshire, U.K.) containing 5% (v/v) defibrinated sheep blood and incubated at 37 °C for 24 h. The colonies were confirmed as *S. suis* by the API 20 Strep kit (BioMérieux, Marcy l’Etoile, France). All of them presented the same resistance pattern as described above.

### 4.2. Essential Oil and Characterization

The essential oil of clove flower bud was purchased from Thai-China Flavours & Fragrances Industry Co. (Thailand). Its chemical characterizations were analyzed using a gas chromatography-mass spectrometry (GC-MS) method. The GC-MS analysis was performed using an Agilent 6890 gas chromatograph in positive ion electron impact (EI, 70 eV) mode coupled to an Agilent 5973 mass selective detector (Agilent Technologies Inc, USA) with an HP-5MS column (30 m × 250 μm i.d. × 0.25 μm film thickness). The individual compound was identified by retention times relative to those of authentic samples and by matching the spectral peaks available in the Wiley, NIST, and NBS mass spectral libraries.

### 4.3. Preparation of Clove Oil Concentration

The clove oil was prepared aseptically in methanol as a 30% (v/v) stock solution [23]. The concentrations used in subsequent experiments were prepared from the stock solution.

### 4.4. Determination of Antibacterial Activity

The antibacterial activity of clove oil was evaluated by disc diffusion method [23]. The MDR *S. suis* strains were inoculated into Bacto™ Todd–Hewitt broth (THB) medium (Becton Dickinson & Co., Sparks, MD, USA) and grown for 18 h at 37 °C. Bacterial cultures were adjusted to a concentration of 1.5 × 10^8^ CFU/mL using a McFarland densitometer (Grant Instruments, Cambridgeshire, UK) and were swabbed on BBL™ Mueller–Hinton agar (MHA) (Becton Dickinson & Co., Sparks, MD, USA) plates containing 5% (v/v) defibrinated sheep blood. Sterile paper discs (Whatman no. 5, 6-mm diameter) were infused with 50 µL of different concentrations of clove oil (1%, 5%, 10%, and 15% [v/v]). The airdried discs were placed on MHA plates and incubated overnight at 37 °C. Penicillin (10 µg) was used as a positive control, and airdried discs of methanol were used as a vehicle control. The zones of inhibition were measured, and the determination was done in three independent experiments.

### 4.5. Minimum Inhibitory Concentration (MIC) and Minimum Bactericidal Concentration (MBC)

MIC and MBC were measured by the broth microdilution method. The inoculums were prepared by suspending bacteria of each isolate in 3 mL of sterile 0.85% NaCl solution. Each suspension was adjusted to 0.5 using a McFarland densitometer and then diluted to 1:200 in BBL™ cation-adjusted Mueller–Hinton broth (CAMHB) (Becton Dickinson & Co., Sparks, MD, USA) supplemented with 5% (v/v) lysed horse blood. The cell density of the final inoculums was 5 × 10^6^ CFU/mL. Equal volumes (100 µL) of bacterial suspension, and two-fold serial dilutions of clove oil in CAMHB supplemented with 5% (v/v) lysed horse blood (0.2–0.0125% final concentration) were mixed into the wells of 96-well plates. Control wells with no bacteria or no clove oil and methanol control were also prepared. After incubation for 24 h at 37 °C, MIC values were recorded as the lowest concentration of clove oil in which no visible growth occurred. To evaluate MBC values, 10 µL of the content from each well showing no visible growth were dropped on MHA plates containing 5% (v/v) defibrinated sheep blood and then incubated for 24 h at 37 °C. The MBCs of clove oil were recorded as the lowest concentration at which no colony formation occurred. The MIC and MBC values were determined in triplicate and in three independent experiments. The MIC and MBC against 50% (MIC50 and MBC50 values) and 90% (MIC90 and MBC90 values) of isolates were also calculated.

### 4.6. Time-Kill Study

The MDR *S. suis* MNCM06 isolate was randomly selected as a representative strain to determine the killing kinetics of clove oil. The overnight culture of bacteria was harvested by centrifugation and suspended in sterile phosphate-buffered saline (PBS; pH 7.2). The suspension was adjusted to a concentration of 1.5 × 10^8^ CFU/mL and treated with clove oil at MBC. A tube without essential clove oil was assigned as the control. Samples were taken at 0, 5, 10, 15, 30, and 60 min, then serially diluted in PBS (pH 7.2) and dropped in triplicate on Columbia blood agar plates containing 5% (v/v) defibrinated sheep blood. Following 24 h of incubation at 37 °C, the colony-forming units (CFU) were calculated. The determination was done in three independent experiments.

### 4.7. Effect of pH on Bactericidal Activity

The effect of pH on the bactericidal activity of clove oil was determined as reported previously with slight modifications [51]. The pH was adjusted by adding 1 M of HCl or 1 M of NaOH to the THB to obtain various pH values (4, 5.5, 7, and 8), sterilized by membrane filtration, and used immediately. The overnight culture of MDR *S. suis* MNCM06 isolate (1.5 × 10^8^ CFU/mL) was inoculated into 5 mL of pH-adjusted media with clove oil at MBC concentration, followed by incubation at 37 °C. Next, 100 μL of the sample was taken, serially diluted in PBS (pH 7.2), and dropped in triplicate on Columbia blood agar plates containing 5% (v/v) defibrinated sheep blood at regular time intervals (0, 15, 30, 60 min). Following 24 h of incubation at 37 °C, the CFU was calculated. All the determinations were done in triplicate.

### 4.8. Scanning Electron Microscopy (SEM)

The effect of clove oil on the morphology of the MNCM06 isolate of MDR *S. suis* was further observed by SEM. Overnight culture of the bacteria was harvested by centrifugation, suspended in PBS (pH 7.2), and then treated with clove oil at MIC and MBC for 1 h at 37 °C and processed as previously described [22]. A control sample without clove oil treatment was similarly prepared and examined.

### 4.9. Statistical Analysis

Data were analyzed and expressed as mean ± SD of experiments performed in triplicate. The data for the effect of pH on bactericidal activity were analyzed using analysis of variance. Pairwise comparisons were analyzed using the least squares means procedure with Tukey’s adjustment (values of *p* < 0.05), and the statistical tests were performed using SAS 9.2 software.

## Figures and Tables

**Figure 1 pathogens-09-00014-f001:**
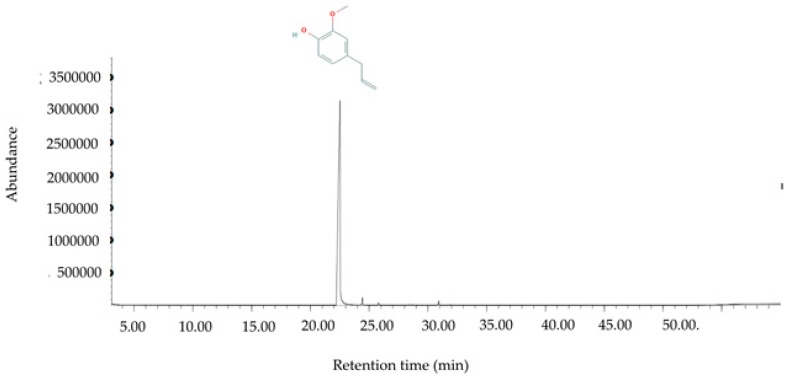
Gas chromatography-mass spectrometry (GC-MS) chromatograph for essential oil of *S. aromaticum* (L.) Merr. & L.M. Perry.

**Figure 2 pathogens-09-00014-f002:**
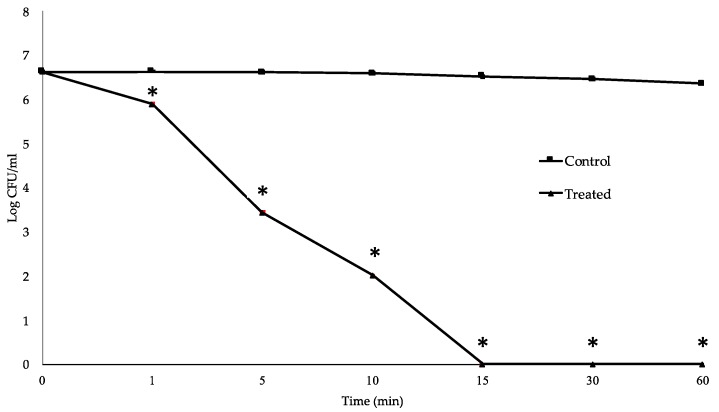
Effect of clove oil on MDR *S. suis* viability. Bacteria were suspended in phosphate-buffered saine (PBS) and incubated with or without oil (control) at MBC. Error bars represent the standard deviations from three independent experiments. * *p* < 0.05.

**Figure 3 pathogens-09-00014-f003:**
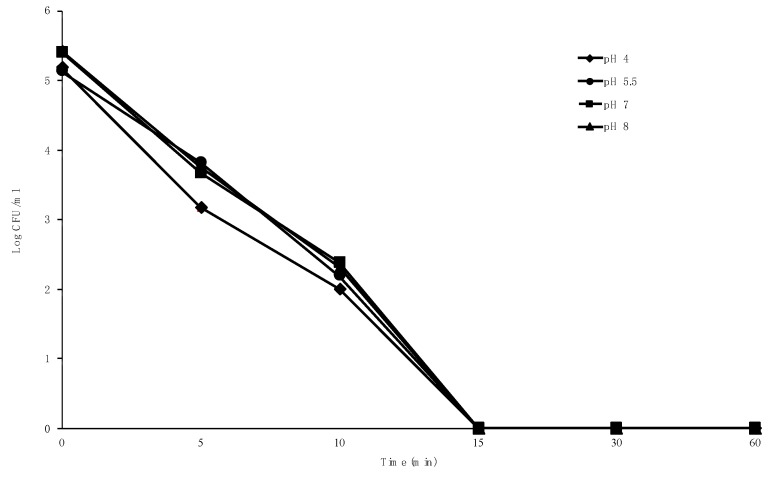
Effect of pH on the bactericidal activity of clove oil against MDR *S. suis*. Error bars represent the standard deviations from three independent experiments.

**Figure 4 pathogens-09-00014-f004:**
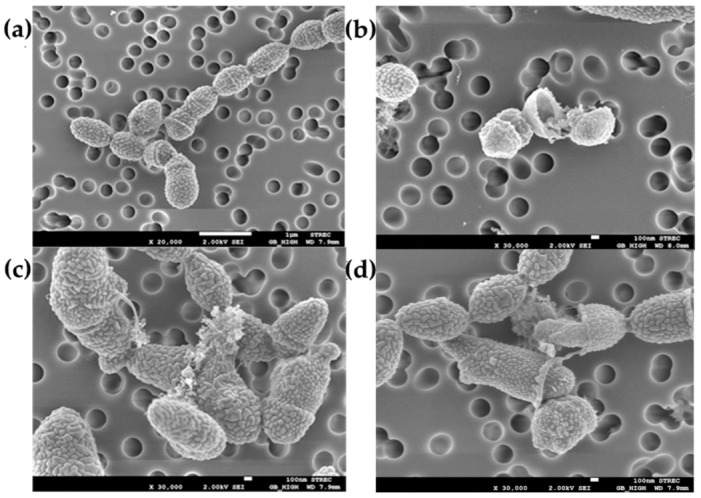
Scanning electron microscopic: (**a**) Image of MDR *S. suis* untreated cell; (**b**) after treatment with clove oil at the MIC, (**c**) and (**d**) after treatment with clove oil at the MBC. Scale: 1 µm (**a**), 100 nm (**b**–**d**).

**Table 1 pathogens-09-00014-t001:** The chemical composition of essential oil from *S. aromaticum* (L.) Merr. & L.M. Perry obtained by GC-MS analysis.

Peak	Retention Time	Phytoconstituents	Molecular Formula	Molecular Weight	Area (%)
1	22.5156	Eugenol	C_10_H_12_O_2_	164.20	97.7573
2	24.4184	Caryophyllene	C_15_H_24_	204.35	1.1647
3	25.7996	Gamma-humulene	C_15_H_24_	204.35	0.3944
4	30.9152	Caryophyllene oxide	C_15_H_24_O	220.35	0.7836

**Table 2 pathogens-09-00014-t002:** Antimicrobial activity of clove oil against multidrug-resistant (MDR) *S. suis*.

*S. suis* Isolates	Inhibition Zone (mm)					MIC (% v/v)	MBC (% v/v)
	Penicillin 10 µg	Clove Oil					
		1%	5%	10%	15%		
PCM01	33.20 ± 0.24	0.00 ± 0.00	0.00 ± 0.00	7.00 ± 0.00	13.00 ± 0.00	0.05 ± 0.00	0.1 ± 0.00
PCM02	15.00 ± 0.00	0.00 ± 0.00	7.13 ± 0.23	10.10 ± 0.08	15.00 ± 0.00	0.10 ± 0.00	0.1 ± 0.00
PCM04	17.30 ± 0.47	0.00 ± 0.00	5.07 ± 0.12	6.03 ± 0.05	8.00 ± 0.00	0.10 ± 0.00	0.1 ± 0.00
PCM06	32.01 ± 0.09	0.00 ± 0.00	8.00 ± 0.00	6.00 ± 0.00	11.07 ± 0.05	0.05 ± 0.00	0.1 ± 0.00
PLP03	15.00 ± 0.00	0.00 ± 0.00	7.00 ± 0.00	7.00 ± 0.00	15.03 ± 0.05	0.05 ± 0.00	0.1 ± 0.00
PLP06	45.00 ± 0.00	0.00 ± 0.00	0.00 ± 0.00	7.07 ± 0.05	14.00 ± 0.00	0.05 ± 0.00	0.1 ± 0.00
PCM05	45.00 ± 0.00	0.00 ± 0.00	0.00 ± 0.00	5.03 ± 0.05	12.10 ± 0.08	0.05 ± 0.00	0.1 ± 0.00
MNCM06	34.01 ± 0.09	0.00 ± 0.00	8.00 ± 0.00	12.00 ± 0.00	15.00 ± 0.00	0.05 ± 0.00	0.1 ± 0.00
MNCM07	35.00 ± 0.00	0.00 ± 0.00	7.23 ± 0.21	10.13 ± 0.09	15.00 ± 0.00	0.05 ± 0.00	0.1 ± 0.00
MNCM10	46.01 ± 0.05	0.00 ± 0.00	13.07 ± 0.12	12.23 ± 0.17	14.00 ± 0.00	0.05 ± 0.00	0.1 ± 0.00
MNCM43	33.00 ± 0.00	0.00 ± 0.00	5.00 ± 0.00	9.93 ± 0.09	10.00 ± 0.00	0.05 ± 0.00	0.1 ± 0.00
MNCM21	35.00 ± 0.00	0.00 ± 0.00	7.00 ± 0.00	9.00 ± 0.00	15.07 ± 0.05	0.05 ± 0.00	0.1 ± 0.00
MNCM25	40.00 ± 0.01	0.00 ± 0.00	8.13 ± 0.12	11.07 ± 0.05	15.00 ± 0.00	0.05 ± 0.00	0.1 ± 0.00
LPH5	39.00 ± 0.02	0.00 ± 0.00	8.00 ± 0.00	14.03 ± 0.05	11.03 ± 0.05	0.05 ± 0.00	0.1 ± 0.00
MNCM50	34.00 ± 0.03	0.00 ± 0.00	7.40 ± 0.35	8.10 ± 0.08	11.00 ± 0.00	0.10 ± 0.00	0.1 ± 0.00

**Table 3 pathogens-09-00014-t003:** Distribution of minimum inhibitory concentrations (MIC) and minimum bactericidal concentrations (MBC) in 15 multidrug-resistant *S. suis* isolates.

Clove Oil Concentration (%v/v)	MIC Distribution by the Number of Isolates	MBC Distribution by the Number of Isolates	MIC50/MBC50	MIC90/MBC90
0.0125	0	0	0.05/0.1	0.1/0.1
0.025	0	0		
0.05	12	0		
0.1	3	15		
0.2	0	0		

**Table 4 pathogens-09-00014-t004:** MDR *S. suis* isolates used in this study.

*S. suis* Isolates	Serotypes **	ST ^†^	Genotype ^‡^	Sources ^§^
			*mrp*/*epf*/*sly*	
PCM01	2	104	+/-/-	healthy pig
PCM02	7	373	+/-/+	healthy pig
PCM04	7	89	-/-/+	healthy pig
PCM06	9	16	-/-/-	healthy pig
PLP03	16	374	-/-/-	healthy pig
PLP06	16	375	-/-/-	healthy pig
PCM05	16	376	-/-/-	healthy pig
MNCM06	2	1	+/+/+	human
MNCM07	14	11	+/*/+	human
MNCM10	2	25	+/-/-	human
MNCM43	2	28	+/-/-	human
MNCM21	2	101	-/-/+	human
MNCM25	2	102	+/-/-	human
LPH5	2	103	+/-/-	human
MNCM50	2	104	-/-/+	human

^**^ Serotypes were identified by multiplex-polymerase chain reaction (PCR) combined with serum agglutination [50]. ^†^ Sequence type (ST) were identified by Multilocus sequence typing (MLST) [50]. ^‡^ Virulence-associated gene profiles were identified by multiplex-PCR; *mrp*, muramidase-released protein; *epf*, extracellular factor; *sly*, suilysin; ^*^ the variant of *epf* [50]. ^§^ All strains were recovered from stock kept at −20 °C.

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
