# Peer review of "Bactericidal Effect of Clove Oil against Multidrug-Resistant Streptococcus suis Isolated from Human Patients and Slaughtered Pigs"

_pathogens, 2019, doi:10.3390/pathogens9010014_

Round 1

Reviewer 1 Report

The manuscript entitled “Bactericidal effect of clove oil against multidrug-resistant Streptococcus suis isolated from human patients and slaughtered pigs in northern Thailand” is well structured and well written.

The topic addressed is topical and of general interest both for the application aspects of essential oils and for the growing phenomenon of antibiotic resistance, which requires that research is directed towards the use of natural molecules.

The abstract is concise but sufficiently exhaustive with regard to the focus elaborated in the work.

The introduction is relevant and well-proportioned; it is also accompanied by an updated bibliography, although some references on the antibacterial activities of the essential oil of cloves could be added.

I suggest some of them here only as an example:

Vieira, B. B., Mafra, J. F., da Rocha Bispo, A. S., Ferreira, M. A., de Lima Silva, F., Rodrigues, A. V. N., & Evangelista-Barreto, N. S. (2019). Combination of chitosan coating and clove essential oil reduces lipid oxidation and microbial growth in frozen stored tambaqui (Colossoma macropomum) fillets. LWT, 116, 108546.

Radünz, M., da Trindade, M. L. M., Camargo, T. M., Radünz, A. L., Borges, C. D., Gandra, E. A., & Helbig, E. (2019). Antimicrobial and antioxidant activity of unencapsulated and encapsulated clove (Syzygium aromaticum, L.) essential oil. Food chemistry, 276, 180-186.

The way in which the results of the GC-MS analysis are tabulated is not too clear. In the meantime I would suggest to insert all the compounds found at least up to those found in percentages higher than 1%. In fact, it is well known that in phytocomplexes the synergistic action of the various components, even minority ones, is often important for antibacterial activity.

A consideration on the synergistic activity of the different molecules should in fact be deepened also during the discussion phase.

However, the materials and methods have been conducted with great scientific rigour and have been well detailed.

The work is very well conducted and in my opinion only needs a few changes and additions to be published.

Author Response

Point 1: The introduction is relevant and well-proportioned; it is also accompanied by an updated bibliography, although some references on the antibacterial activities of the essential oil of cloves could be added. I suggest some of them here only as an example:

Vieira, B. B., Mafra, J. F., da Rocha Bispo, A. S., Ferreira, M. A., de Lima Silva, F., Rodrigues, A. V. N., & Evangelista-Barreto, N. S. (2019). Combination of chitosan coating and clove essential oil reduces lipid oxidation and microbial growth in frozen stored tambaqui (Colossoma macropomum) fillets. LWT, 116, 108546.

Radünz, M., da Trindade, M. L. M., Camargo, T. M., Radünz, A. L., Borges, C. D., Gandra, E. A., & Helbig, E. (2019). Antimicrobial and antioxidant activity of unencapsulated and encapsulated clove (Syzygium aromaticum, L.) essential oil. Food chemistry, 276, 180-186.

Response 1: The references are added and highlighted in text of introduction (line 69) and reference (lines 389-396) section. [Edit with attached file]

Point 2: The way in which the results of the GC-MS analysis are tabulated is not too clear. In the meantime, I would suggest inserting all the compounds found at least up to those found in percentages higher than 1%. In fact, it is well known that in phytocomplexes the synergistic action of the various components, even minority ones, is often important for antibacterial activity.

Response 2: I particularly agree in your suggestion that the efficacy of herb result from synergistic effect of the phytocomplexes. However, the clove oil in this study contains only four chemical compositions including Eugenol (97.75), Caryophyllene (1.16), Gamma-humulene (0.39) Caryophyllene oxide (0.78).

Point 3: A consideration on the synergistic activity of the different molecules should in fact be deepened also during the discussion phase.

Response 3: Discussion of the synergistic activity of the different molecules is added in the first paragraph (lines 150-153) of discussion section. [Edit with attached file]

Reviewer 2 Report

The Authors decribed a manuscript on the antimicrobial effect of clove oil on MDR S. suis. Moreover, the Authors investigated the content of clove oil with mass spectrometry. The results are supported by presented data, however the originality of the study is limited due to the presence of previously published similar studies.

Major concerns

Lines 72-73. The Authors underlined that this is the first study to describe the antimicrobial effect of clove oil on suis isolates, however a similar study has been recently published (Aguiar et al. MicrobiologyOpen 2018). The original aspect is that clove oil has been studied on MDR S. suis strains in the current manuscript. Moreover, as reported, also the mass spectrometry of clove oil has been previously determined. The Authors should underline these limitations in the discussion section.

Minor concerns

Line 59-60. Please change “this could lead to unsuccessful in the treatment” to “this could lead to an unsuccessful treatment”. Line 63. Please change “investigation” to “investigations”. suis should be in italics, please correct in the text. Line 178. “Sensitive” should be replaced with “susceptible”. Line 207. “in vivo” should be in italics.

Author Response

Point 1: Lines 72-73. The Authors underlined that this is the first study to describe the antimicrobial effect of clove oil on S. suis isolates, however a similar study has been recently published (Aguiar et al. MicrobiologyOpen 2018). The original aspect is that clove oil has been studied on MDR S. suis strains in the current manuscript. Moreover, as reported, also the mass spectrometry of clove oil has been previously determined. The Authors should underline these limitations in the discussion section.

Response 1: The sentence in lines 70-73 has revised to “Although the antibacterial properties of clove oil have been previously reported, however, the study of its effect on MDR S. suis has limitedly examined. The aims of this study were to determine the chemical composition of essential oil from clove buds and to investigate the antibacterial effects of clove oils on MDR S. suis.” [Edit with attached file]

The sentence in lines 160-164 of discussion section has also revised to “Even though the antibacterial properties of essential oil from clove bud against S. suis have already been reported, its effect against MDR S. suis has not been previously reported. Our findings suggest that the MDR S. suis isolated from both human patients and slaughtered pigs is susceptible to essential oil of S. aromaticum (L.) Merr. & L.M. Perry." [Edit with attached file]

Point 2: Line 59-60. Please change “this could lead to unsuccessful in the treatment” to “this could lead to an unsuccessful treatment”. Line 63. Please change “investigation” to “investigations” S. suis should be in italics, please correct in the text. Line 178. “Sensitive” should be replaced with “susceptible”. Line 207. “in vivo” should be in italics.

Response 2: Thank you for your kindness. The sentence “this could lead to unsuccessful in the treatment” has reformulated and highlighted (line 58). The word “investigation” is corrected and highlighted (line 61), “S. suis” is corrected in italics, “Sensitive” is replaced with “susceptible” (line 180), and “in vivo” is corrected (line 209). [Edit with attached file]

Reviewer 3 Report

The authors confirmed eugenol as the major component of clove oil and investigated the potential of clove oil to act as an antimicrobial against MDR S. suis. The authors could show that 15 S. suis isolates representing different sequence types and serotypes are susceptible to clove oil at 0.1%. Considering emerging antibiotic resistance of S. suis especially in Thailand, this finding is of relevance to the field and points to a potential use of clove oil in the treatment of S. suis disease.

The study is well designed and the results are presented in an intelligible manner. The authors used both the disc diffusion and the broth microdilution method to determine the effect of clove oil on S. suis growth and viability.

However, my main concern is the use of the English language. Grammar mistakes were omnipresent throughout the manuscript. The manuscript cannot be accepted for publication unless the quality of the writing is improved. A native speaker or professional interpreter should be consulted.

Furthermore, I have some minor concerns:

In lines 58-59, the authors state that S. suis is a newly emerging MDR zoonotic pathogen. Where is the reference for this statement?

The resolution of Fig. 1 could be improved.

Concerning Figure 2, why is the starting inoculum (timepoint 0) already almost one log less for treated S. suis compared to control?

Was there a statistically significant difference at any timepoint for growth of clove oil treated S. suis versus control? If so, please indicate this in the Figure 2.

What are the small dots in the background of each picture in Fig. 4?

Figure 4: Considering the pronounced bactericidal effect of clove oil shown in Table 2 and Figure 2, I am surprised the morphological changes observed by SEM were not more drastic. The majority of S. suis in Fig. 4 c and d still look intact. I suggest the authors quantify the % of cells with abnormal morphology and compare it to untreated cells.

Author Response

Point 1: In lines 58-59, the authors state that S. suis is a newly emerging MDR zoonotic pathogen. Where is the reference for this statement?

Response 1: The references are added and highlighted (lines 57 and 364-371) . [Edit with attached file]

Point 2: The resolution of Fig. 1 could be improved.
Response2: The GC-MS chromatograph in Fig. 1 has been improved the resolution. [Edit with

attached file]

Point 3: Concerning Figure 2, why is the starting inoculum (timepoint 0) already almost one log less for treated S. suis compared to control? Was there a statistically significant difference at any timepoint for growth of clove oil treated S. suis versus control? If so, please indicate this in the Figure 2.

Response 3: Thank you for your question. In fact, the initial inoculums composed of bacterial suspensions (4.5 x 106 cfu/ml) in both control and treatment groups. The result illustrated that the clove immediately killed S. suis within 1 minute. The time-killing analysis in Fig. 2 has been edited. [Edit with attached file]

Point 4: What are the small dots in the background of each picture in Fig. 4?
Response 4: The small dots in the background is the pore of membrane filters (isoporeTM,

Merck). This membrane filter was used in the method of sample preparation for SEM.

Point 5: Figure 4: Considering the pronounced bactericidal effect of clove oil shown in Table 2 and Figure 2, I am surprised the morphological changes observed by SEM were not more drastic. The majority of suis in Fig. 4 c and d still look intact. I suggest the authors quantify the % of cells with abnormal morphology and compare it to untreated cells.

Response 5: Because of reducing the bias on a personal evaluation, the scanning electron microscope photographs was used for qualitative research. The treated cells present particularly cellular deforming, lysis of cell wall, and losing of cytoplasmic contents (Fig. 4 c and d) whereas, untreated cells is typical morphology (Fig. 4a). However, quantitative antibacterial efficacy of the clove oil has been demonstrated in the disc diffusion method and the broth microdilution method also.

Remark: The reviewer concerns the use of the English language

• The English grammar is during revision.

Round 2

Reviewer 2 Report

The Authors have successfully addressed all reviewers suggestions.

Author Response

Dear Reviewer

Thank you very much for your message. We have studied on reviewer comments carefully and made major correction. The english language and style have also edited under the guidance of the academic editor already.

If any further information is required, please do not hesitate to contact me. Thank you for your kindness.

Sincerely Yours,

Raktham Mektrirat, D.V.M., Ph.D. (Corresponding author)